# Nuclear Morphological Remodeling in Human Granulocytes Is Linked to Prenylation Independently from Cytoskeleton

**DOI:** 10.3390/cells9112509

**Published:** 2020-11-20

**Authors:** Sebastian Martewicz, Camilla Luni, Xi Zhu, Meihua Cui, Manli Hu, Siqi Qu, Damiano Buratto, Guang Yang, Eleonora Grespan, Nicola Elvassore

**Affiliations:** 1Shanghai Institute for Advanced Immunochemical Studies (SIAIS), ShanghaiTech University, Shanghai 201210, China; smartewicz@shanghaitech.edu.cn (S.M.); camilla.luni@shanghaitech.edu.cn (C.L.); zhuxi1@shanghaitech.edu.cn (X.Z.); cuimh@shanghaitech.edu.cn (M.C.); huml@shanghaitech.edu.cn (M.H.); qusq@shanghaitech.edu.cn (S.Q.); dburatto@shanghaitech.edu.cn (D.B.); yangguang@shanghaitech.edu.cn (G.Y.); 2School of Life Science and Technology, ShanghaiTech University, Shanghai 201210, China; 3Institute of Biochemistry and Cell Biology, Shanghai Institutes for Biological Sciences, Chinese Academy of Sciences, Shanghai 200031, China; 4Institute of Neuroscience, National Research Council, 35127 Padova, Italy; eleonora.grespan@gmail.com; 5Department of Industrial Engineering, University of Padova, 35131 Padova, Italy; 6Venetian Institute of Molecular Medicine, 35129 Padova, Italy; 7Stem Cells & Regenerative Medicine Section, UCL Great Ormond Street Institute of Child Health, London WC1N 1EH, UK

**Keywords:** nuclear morphology, granulocyte lobulation, nuclear segmentation, prenylation, GGTase III

## Abstract

Nuclear shape modulates cell behavior and function, while aberrant nuclear morphologies correlate with pathological phenotype severity. Nevertheless, functions of specific nuclear morphological features and underlying molecular mechanisms remain poorly understood. Here, we investigate a nucleus-intrinsic mechanism driving nuclear lobulation and segmentation concurrent with granulocyte specification, independently from extracellular forces and cytosolic cytoskeleton contributions. Transcriptomic regulation of cholesterol biosynthesis is equally concurrent with nuclear remodeling. Its putative role as a regulatory element is supported by morphological aberrations observed upon pharmacological impairment of several enzymatic steps of the pathway, most prominently the sterol ∆14-reductase activity of laminB-receptor and protein prenylation. Thus, we support the hypothesis of a nuclear-intrinsic mechanism for nuclear shape control with the putative involvement of the recently discovered GGTase III complex. Such process could be independent from or complementary to the better studied cytoskeleton-based nuclear remodeling essential for cell migration in both physiological and pathological contexts such as immune system function and cancer metastasis.

## 1. Introduction

In recent years, nuclear shape is transcending the definition of mere morphological cell feature and is increasingly recognized for its structural purposes and its role in regulating cellular functions. Indeed, control over nuclear remodeling is essential for cell migration [1], regulates gene transcription [2], and aberrant nuclear morphologies are one of the hallmarks of cancers, scaling with malignancy [3].

Nevertheless, the physiological variety of shapes and structures assumed by the nucleus often remains orphan of a clear molecular mechanism of formation. For instance, the nuclear tunnels and invaginations observed in most in vitro-studied cells [4] increase nucleus-cytoplasm interface area and facilitate transport of ions and RNA [5,6,7], but their assembly method remains matter of intense speculation [8,9]. Similarly, nuclear segmentation of granulocytic nuclei represents a structural remodeling resilient to harsh dynamic deformations involved in migration [10] and explosive chromatin decondensation in NETosis [11], with only a putative involvement of cytoskeletal forces for its progression [12].

Indeed, the cytoskeleton is broadly regarded as the main regulator of nuclear morphology, especially as a consequence of studies in migratory cells, where dynamic nuclear remodeling relies on specialized cytoskeletal structures exerting forces to push, pull, and squeeze while pivoting on cell-substrate anchoring points [1,13,14,15,16]. In such systems, nuclear morphology is directly linked to extracellular features. Nevertheless, the molecular mechanism for the generation and maintenance of nuclear invaginations, tunnels and segments are not clearly overlapping with those necessary for nuclear remodeling during cell migration. While cytoplasmic cytoskeletal elements coexist with these structures [17], the causality between the two is still debated [9] and thus far only observed under special or pathological conditions [18,19].

A nucleus-autonomous mechanism for controlling nuclear morphology is a compelling hypothesis arguing for the involvement of either molecular machinery dedicated to structuring the NE or chromatin-associated pulling forces [8,9]. Indeed, several components of the endosomal machinery are equally able to bind, bend, and fuse nuclear membranes [20,21] and regulate NE integrity [22]. Moreover, DNA filaments and NE tethering possess impressive tensile strengths capable of deforming nuclei across hundreds of microns when pulled by migrating cells [23] and chromatin dynamic reorganization is the motor behind nuclear invaginations in *Drosophila melanogaster* polytene nuclei [24]. Nuclear segmentation concurrent to granulocytic differentiation is equally associated with chromatin reorganization, with deposition of nuclear envelope-limited chromatin sheets at NE bending points in human cells [25,26] and wide-spread chromosomal supercontraction in murine cells [27].

In this study, we exploit the suspended nature of myeloid cells to isolate the cellular system from extracellular forces and substrate-anchoring points, and we take lobulation and segmentation of granulocyte nuclei as a model for cell-intrinsic nuclear remodeling.

In vivo, remodeling of the spherical myeloid nucleus is a three-stage process across bean-shaped nuclei in metamyelocytes, proto-lobulation in band cells and final nuclear segmentation in granulocytes when nuclear lobules separate, linked by thin DNA-containing filaments [28,29]. Here, we show that cytosolic cytoskeleton does not contribute to maintenance or generation of nuclear lobules and nuclear segments.

In vivo, differentiation is uncoupled from nuclear remodeling, as shown by functionally mature granulocytes displaying round or non-lobulated nuclei upon mutations in laminB-receptor (LBR) gene [30,31,32]. Given this concurrent but not necessarily causative relationship, we temporally profile transcriptomic changes in differentiating granulocytes and identify a metabolic pathway involving the enzymatic activity of LBR as temporally concurrent with nuclear remodeling. Ultimately, targeted biochemical challenging of several enzymes participating in this pathway reveals a putative contribution of the enzymatic activity of LBR in nuclear lobulation and the essential role of protein prenylation in both lobulation and nuclear segmentation.

## 2. Materials and Methods

All experimental procedures are further detailed in the Extended Materials and Methods section in the Appendix A.

### 2.1. Cell Cultures

HL60 cells were from ECACC (Sigma-Aldrich, St. Louis, MI, USA, cat#98070106) and maintained in RPMI 1640 (Thermo Fisher Scientific, Waltham, MA, USA) + 10% FBS (Thermo Fisher Scientific). Granulocytic differentiation was induced by 5 μM all-trans-retinoic acid (Sigma-Aldrich, St. Louis, MO, USA) at Day 0 to 2 × 105 cell/mL cultures. For RNA collection, at Day 2 iHL60 cultures were diluted 1:5 with fresh medium. Biological replicates are independent differentiation protocols of subsequent culture passages.

### 2.2. RNA Processing

Total RNA was isolated at 0, 48, and 96 h of ATRA treatment from 10^7^ cells with TRIzol Reagent (Thermo Fisher Scientific) followed by purification with RNeasy Mini Kit (Qiagen, Hilden, Germany). 5 μg of total RNA were further processed at GeneWiz, Suzhou, China. For real time PCR, High Capacity cDNA Reverse Transcription kit (Thermo Fisher Scientific) and PowerUp™ SYBR™ Green Master Mix (Thermo Fisher Scientific) were used. Primer sequences are reported in Appendix A.

### 2.3. Bioinformatics Analyses

RNA data were processed as previously reported [33]. For updated software versions and detailed description of data filtering, see Supplementary Information. Gene expression data are publicly available on Gene Expression Omnibus database (http://www.ncbi.nlm.nih.gov/geo) under the GEO IDs: GSE134922.

### 2.4. Drug Treatments

Targets, suppliers and references for each drug are reported in Appendix A. Length of treatment and drug concentration vary and are reported in the text. In double treatment experiments, all compounds were administered simultaneously, with the exception of 3-day long experiments, where cells were pre-treated for 1 h with either latrunculin A or Y-27632 before vincristine sulfate supplementation.

### 2.5. Live-Cell Imaging

Cell nuclei were stained with 1 μg/mL Hoechst 33,342 (Cell Signaling Technologies, Danvers, MA, USA). The endoplasmic reticulum was stained with 2 mM ER-Tracker™ Blue-White DPX (Thermo Fisher Scientific). Imaging was performed with an inverted Zeiss LSM710 laser-scanning confocal microscope, 100× oil-immersion objective, 405 nm excitation wavelength and a 0.5 μm step.

### 2.6. Image Analyses

For volume and surface quantifications, images of nuclei stained with ER-Tracker™ were processed with the Image Processing Toolbox of MATLAB software (R2015b).

### 2.7. Qualitative Evaluation of Nuclear Lobulation

The “Number of lobules” was manually derived for each nucleus from Hoechst33342 staining images and plotted as count distribution for number of lobules. The “Maximum number of sections” was manually derived from ER-Tracker staining images by considering the maximum number of nuclear sections in a cell for any given focal plane in the z-stack, and plotted as count distribution for number of sections. For qualitative analyses, the three categories were defined as “Round/Ovoid”, according to geometry, “Segmented” if the nucleus presented at least 2 well defined separated volumes, and “Deformed” when neither of the previous two applied. Qualitative evaluation is presented as percentage on total population for each class.

### 2.8. Flow Cytometry Analyses

Antibody staining was performed according to manufacturer’s suggestions. All antibodies were from Biolegend, San Diego, CA, USA: CD11b (cat#301309), CD54 (cat#353105), CD62L (cat#304803). Respiratory burst assay was performed according to manufacturer’s instructions (Cayman Chemical Company, Ann Arbor, MI, USA, cat#601130). Samples were analyzed with the CytoFLEX Platform (Beckman Coulter, Pasadena, CA, USA) analyzing at least 10^4^ single cell events.

### 2.9. Molecular Modeling

The models for the FTase, GGTase I, GGTase II, and GGTase III were derived from published crystal structures (Protein Data Bank (PDB) ID codes 1S63, 1S64, 3DSS and 6J74/6J7X respectively). Missing residues were modeled using the MODELLER package. GGTaseII complexed with two PO_4_^3−^ ions was modelled starting from the crystal structure of the complex GGTase II-GGPP (PDB ID: 3DST). Atomic coordinates for the 3D structure of the L-778,123 were obtained starting from the smile code using ChemOffice18 packaged. Docking has been performed using AutoDock4ZN package.

### 2.10. Statistical Analyses

Data for cell and nuclear areas and volumes are presented as median and interquartile range (IQR). Lognormal data distribution was normalized and statistical analyses were performed by one-way ANOVA followed by Tukey–Kramer test in computing environment R. Data for lobule and section numbers are presented as lognormal distributions of counts per cell normalized for width. Statistics were performed by non-parametric Kruskal–Wallis test with post-hoc Dunn test for pairs. Data plotting and statistics were performed with Origin 2020 software.

## 3. Results

### 3.1. Lobular Structures in Segmented Nuclei Are Not Maintained by the Cytoskeleton

We first sought to define the role of the three main cytoskeletal components (actin microfilaments (MFs), microtubules (MTs), and intermediate filaments (IFs)) in maintaining the lobular and segmented nuclear structure in induced HL60 cells (iHL60). All three networks can be biochemically targeted in a specific manner (Figure 1a and Appendix A).

We treated Day 3 iHL60 for 3 h with each compound, quantifying confocal cell z-stacks for two parameters: Number of lobules distinguishable after live-cell DNA staining (Figure 1b) and maximum number of nuclear sections visible on one focal plane after live-cell endoplasmic reticulum (ER) staining (Appendix A). At Day 3, iHL60 present a segmented nucleus with a median of 3 lobules per nucleus (Figure 1c). MT depolymerization with vincristine (VNCT, Figure 1d) or nocodazole (Noco, Appendix A) abolishes lobular structure, although nuclear invaginations and segmentation are retained in the newly “wrinkled” morphology, as shown by ER intrusions in the nuclear volume (Appendix A).

Interestingly, iHL60 nuclei undergo an almost opposite morphological change following MF stabilization with jasplakinolide (Jasp, Figure 1e), generating extremely hyper-segmented nuclei with well-defined lobules or a morphology we define as “choked” (Appendix A). On the contrary, neither MF depolymerization with latrunculin A (LatA, Figure 1f) or cytochalasin D (CytD, Appendix A) nor MT stabilization with paclitaxel (Pax, Appendix A) affect lobular structure or segmentation (Figure 1b and Appendix A). Equally, IF-disruptors acrylamide, iminodipropionitrile (AA and IDPN, Appendix A) and withaferin A (WTA, Figure 1g) do not interfere with lobular structure or segmentation, aside for an apparent increase in inter-lobule distance in WTA-treated cells. In order to ensure that the nuclear morphology is in steady state condition, we extended the treatment to 24 h between Day 3 and Day 4. The 24-h treatments closely reproduce the results obtained by 3-h treatments, confirming the results and ensuring that neither “wrinkled” nor “chocked” nuclear morphologies characterize apoptotic progression (Appendix A).

A sudden unbalance in cytoskeletal forces could play a role in driving the appearance of both “wrinkled” and “choked” morphologies. To test this hypothesis, we set up double-treatment experiments targeting multiple cytoskeletal components or actomyosin contraction. Strikingly, MF depolymerization completely rescues VNCT-induced nuclear “wrinkling” (Figure 1h). Similarly, lobular structures are rescued from both VNCT- and Jasp-induced deformation by inhibiting either actomyosin contraction with blebbistatin (Blebb, Appendix A) or RhoA-effector kinase with Y-27632 (Figure 1i,j), with the latter displaying a more robust effect. These results provide clear evidence of abnormal actomyosin hypercontraction as main contributor to morphological changes associated with both “wrinkled” and “chocked” nuclei, rather than dependence of lobular structure on MT-network integrity (Figure 1k). Additionally, this effect is specific for myosin-II complexes, as inhibitors of other myosin types (myosin-I, -V, -VI) fail to rescue nuclear “wrinkling” (Appendix A). Surprisingly, while double treatment with IF-disruptors and VNCT does not produce morphological rescues (Appendix A), all three drugs appear to rescue Jasp-induced nuclear “chocking” to varying degrees, with acrylamide showing the strongest effect (Appendix A).

In summary, our experiments show that maintenance of lobular structures and nuclear segmentation in iHL60 is independent from cytoplasmic cytoskeleton, although cytoskeletal forces clearly influence nuclear morphology in general. It is important to highlight how, while single cytoskeleton networks can be targeted specifically, the complex network-interactions of MFs, MTs, and IFs make targeting them independently a challenging endeavor, as shown by MT-mediated regulation of actomyosin contraction and an undefined role of IF-disruptors in MF stabilization.

### 3.2. Nuclear Lobulation and Segmentation Progresses without the Involvement of Cytoskeleton

We then focused on defining the role of MF- and MT-networks in the generation of lobular structures and nuclear segmentation. Long-term exposure of HL60 cells to LatA is lethal, but 3 days of MF-depolymerizing treatment generates giant multinucleated cells, indicative of cycling cells failing cytokinesis (Appendix A). Inhibition of actomyosin contraction generates similar phenotypes, while inhibition of ROCK does not impair cell proliferation/viability or cell division (Appendix A). Taking advantage of this viability window, we performed iHL60 differentiation experiments in presence of LatA, Blebb, and Y27632. Both cytokinesis-impairing drugs LatA and Blebb generate giant multinucleated cells with clearly defined lobular structures and segmentation (Figure 2a), indicating that actomyosin contraction is dispensable from this specific nuclear remodeling process. Inhibition of ROCK produces little effect on lobulation and nuclear segmentation (Figure 2b), with occasional appearance of medium-sized cells (Appendix A).

Equivalent experiments with MT-depolymerizing compounds are more challenging, as undifferentiated HL60 cells are extremely sensitive to even nanomolar VNCT concentrations that lead to mitotic blockade followed by apoptosis. Similarly, viability of induced iHL60 differentiated in presence of VNCT is reduced to ~3% above 1.25 nM (Appendix A). Surviving cells display huge nuclei suggesting failed nuclear divisions as expected in cells with compromised MT-networks, with a roughly equal number of lobulated/segmented nuclei (Figure 2c, left) and “wrinkled” ones, still clearly displaying deep nuclear folds (Figure 2c, right). Simultaneous disruption of both MT- and MF-networks generates both multi- and mono-nucleated cells with abnormally large lobulated and segmented nuclei (Appendix A).

While such low drug concentrations are clearly enough to impair mitotic spindle function, MT-contribution to nuclear remodeling might not be fully lost, thus we sought additional evidence by postponing treatment to Day 1. At Day 1 of differentiation, iHL60 display mostly deformed bean-shaped nuclei without marked lobulation or segmentation (Appendix A) and cell proliferation is slowing down as a consequence of terminal differentiation and cell-cycle exit (Appendix A). A significant proportion of iHL60 cells treated at this time-point for 48 h is resistant to much higher VNCT concentrations with viability above 50% at 12.5 nM, with lobulated and segmented fused nuclei (Appendix A). “Wrinkled” morphology is apparent above 25 nM, with deep invaginations within the nuclear volume and nuclear in-folding (Figure 2d). Similarly to our previous approach, co-treatment with LatA rescues nuclear “wrinkling”, generating enlarged lobulated and segmented nuclei across all VNCT concentrations (Figure 2d and Appendix A). Treatment with Y27632 was effective in rescuing nuclear wrinkling at 25 nM VNCT, with both lobulation and segmentation apparent (Figure 2d and Appendix A).

In summary, our results show that both MF- and MT-network integrity is dispensable for nuclear lobulation and segmentation. Moreover, the cellular phenotypes observed strongly suggested that major cellular events such as cellular and even nuclear divisions do not contribute to the nuclear remodeling process, as both multinucleated cells and polyploid nuclei undergo lobulation and segmentation.

### 3.3. Transcriptomic Regulation of Sterol Metabolism Is Concurrent with Nuclear Remodeling

In order to gain deeper insight into cellular processes concurrent with granulocytic differentiation, we turned to transcriptomic profiling. We considered the nuclear and cellular morphological changes over 4 days after all-trans-retinoic acid (ATRA) supplementation, defining three time-points of interest by considering Day 2 a transition point significantly different from Day 0 and Day 4 for several nuclear parameters (Appendix A).

First, we considered in a targeted manner the cellular components (CCs) relative to MT- (Appendix A and Dataset 1) and MF-cytoskeletons (Appendix A and Dataset 2). Reactome categories enriched in differentially expressed genes (DEGs) in these CCs show once again changes to the cytoskeleton related to differentiation, without clear indication for its involvement in nuclear remodeling, with cell cycle exit and immune system processes as most prominent categories for both MTs and MFs. In our data, IF cytoskeleton is represented by 7 out of over 70 genes codifying for IF-proteins, only 4 of which expressed at significant levels in iHL60 including Type V lamins, which are downregulated components of the inner nuclear envelope (Appendix A). Thus, we considered the nuclear envelope CC in more detail (Figure 3a and Dataset 3). While several categories again are linked to the differentiation process, such as cell-cycle and RNA nuclear export within the downregulated DEGs, among the upregulated genes several categories refer to lipid and specifically sterol metabolism. Intriguingly, we obtained similar results by analyzing whole transcriptomic data in a category-unbiased fashion.

Whole transcriptomic profiles precisely cluster all samples according to time-point (Appendix A), with up- and down-regulated genes evenly distributed in all comparisons (Appendix A and Dataset 4) and the majority of DEGs observed already between Day 2 and Day 0 (Appendix A). Reactome analyses of DEGs between Day 4 and Day 0 confirm successful induction towards terminally differentiated granulocytes (Figure 3b and Dataset 5). By accounting for all three time-points, we clustered DEGs according to their temporal dynamic over 4 days (Figure 3c and Dataset 6). While clusters of upregulated (either definitively or transiently) DEGs mainly confirm the specificity of HL60 differentiation towards granulocytes/neutrophils (Dataset 6), clusters of downregulated DEGs provide an interesting picture for temporal gene downregulation sequence, with an initial phase of ribosomal and nucleolar downregulation matching the reduction in total RNA content per cell (Appendix A), and a second phase of proliferation arrest (Appendix A) and granulocytic maturation (Appendix A) concurrent with nuclear segmentation (Appendix A). Interestingly, once again we observed enrichment in DEGs relating to lipid metabolism and specifically cholesterol biosynthesis (Figure 3d and Appendix A), which prompted us to look in more detail into this category. While three nodes of this gene network display both up- and down-regulated genes, the “cholesterol biosynthesis” node is homogeneously downregulated (Figure 3e), with the notable exception of laminB-receptor (LBR).

In summary, by considering nuclear remodeling as a concurrent but independent process to granulocytic differentiation, we identified a metabolic pathway which temporal transcriptomic regulation correlates with nuclear lobulation and segmentation. The known involvement of one pathway component (LBR) in nuclear remodeling in granulocytes prompted further investigation of this metabolic pathway.

### 3.4. Intermediates of the Cholesterol Pathway and the Isoprenoid Branch Impact Nuclear Remodeling

The cholesterol biosynthetic pathway functionally divides into three parts, with the mevalonate pathway splitting into isoprenoid and post-squalene branches after farnesyl pyrophosphate (FPP) synthesis (Figure 4a). Most genes of participating enzymes show significant differential expression, regardless of fold change (at *p* < 0.001, Figure 4a and Appendix A thick border arrows, Dataset 1). Interestingly, while the post-squalene branch is ultimately heavily downregulated at Day 4, the isoprenoid branch is upregulated at Day 4, following a transient but significant upregulation of enzymes just upstream the branching point at Day 2.

While the transcriptomic regulation of this metabolic pathway might be merely concurrent with differentiation, we pharmacologically targeted several enzymatic steps (red crosses Figure 4a and Appendix A) to test any involvement in nuclear remodeling. Inhibition of enzymatic activity of downregulated elements of the mevalonate pathway and post-squalene branch at any sub-lethal concentration affected neither nuclear lobulation nor segmentation both according to quantification of morphometric parameters (Figure 4b,c) and morphological evaluation (Appendix A). Interestingly, inhibition of ∆14-sterol reductase activity with AY9944 generated a significant population of hypolobulated iHL60 (Figure 4d and Appendix A) displaying a range of aberrant morphologies, including bean-shaped and not-segmented lobulated ones (Movie S1).

The most striking effects on nuclear morphology are apparent upon inhibiting protein prenylation, one end-point process of the isoprenoid branch. Individual inhibition of farnesyltransferase (FTase) or type 1 geranylgeranyltransferase (GGTase I) produces opposite effects with significantly hypo- and hyper-lobulated nuclei, respectively (Figure 4d and Appendix A). FTase and GGTase I are known to display compensatory effects over protein prenylation, thus we sought to inhibit both enzymatic activities with dual inhibitor L-778,123, which leads to complete abrogation of nuclear lobulation and segmentation (Figure 4e,f). The resulting cell population is almost entirely depleted of segmented cells (Figure 4g), with significant fractions displaying qualitatively round or ovoid shapes, quantifiably different by nuclear morphometric parameters (Figure 4h).

In summary, our results show that cholesterol biosynthesis in general is not involved in remodeling, although specific inhibition of TM7SF2/LBR enzymatic activity induces hypolobulated phenotypes. Additionally, the end-step of the isoprenoid branch of protein prenylation produces interesting phenotypes with complete lobulation and segmentation abrogation in L-778,123-treated iHL60 cells.

### 3.5. Protein Prenylation Is Essential for Nuclear Remodeling

GGTase I inhibition with GGTI-298 interferes with cellular division, generating multinucleated HL60 cells (Appendix A) and giving rise to apparently hyperlobulated iHL60 nuclei (Movie S2) in a similar fashion to cells differentiated following MF depolymerization (Figure 2a). On the other hand, FTase inhibition with FTI-277 smoothens HL60 nuclei to an almost spherical shape (Appendix A) and impairs the nuclear segmentation step in iHL60, as nuclear lobules are still present but with a significant fraction displaying lobules connected by wide nuclear constrictions unlike normal nuclear filaments (Movie S3). Most interestingly, dual inhibition of both FTase and GGTase I with L-778,123 reproduces the smoothening effect on HL60 nuclei observed in FTI-277-treated cells (Appendix A), but produces the most dramatic effect on iHL60, generating significant populations of ovoid nuclei and almost entirely abrogating both lobulation and segmentation (Movie S4).

We first sought to verify that the effect on nuclear morphology of L-778,123 is independent from the differentiation process. 72-h treatment of undifferentiated HL60 cells stops proliferation but does not impair viability or differentiation potential (Figure 5a). iHL60 differentiation in presence of L-778,123 leads to strong upregulation of granulocyte/neutrophil markers ITGAM (CD11b), JAML (AMICA1), and ITGB2 (CD18) (Figure 5b), as well as consistent up-, down- or non-regulation of a wider panel of genes compared with transcriptomic data, with strong upregulation of all immune system-related transcripts (Appendix A). Nevertheless, inhibition of prenylation impairs cell membrane presentation of surface markers CD11b, CD54, and CD62L (Appendix A) and respiratory burst capacity (Appendix A).

Surface antigen presentation is reasonably impaired upon L-778,123 treatment, given the involvement of protein prenylation in vesicle trafficking and cell membrane fusion. Thus, we sought to define the time-window of L-778,123 effectiveness in abolishing nuclear lobulation without impairing antigen presentation at later stages by designing drug add-in and washout experiments. Treatment after Day 1 of ATRA induction does not significantly affect neither morphometric parameter tested (Appendix A). Washout experiments identify a similar time-frame, with 24-h ATRA treatment already sufficient to induce robust nuclear lobulation and segmentation at Day 3, which is abrogated by concurrent L-778,123 treatment (Appendix A). Limiting iHL60 treatment to the first 24 h of ATRA-induction generates similar nuclear morphology distributions as 72-h treatments (Figure 5c–e), while simultaneously allowing membrane localization of granulocytic/neutrophil markers and a functional respiratory burst (Figure 5f), thus confirming L-778,123 effect on nuclear morphology is mechanistically independent from differentiation processes.

Specific inhibition of GGTase I does not impair nuclear remodeling, while specific inhibition of FTase even at high drug concentrations does not reproduce the morphology observed with the dual FTase/GGTase I inhibitor-treatment. Thus, we tested if simultaneous specific inhibition is required for lobulation- and segmentation-abrogation. Surprisingly, iHL60 cells treated with a combination of FTI-277 and GGTI-298 do not present the same phenotype observed upon dual inhibition with L-778,123, instead displaying a combination of phenotypes observed in single drug treatments (Figure 5g,h and Appendix A) with hypolobulated and multinucleated cells. The resulting nuclei are not significantly different from FTI-277 treatment (Figure 5i), suggesting FTase activity is the main contributor to this hypolobulated phenotype, but qualitatively and quantitatively different from L-778,123 treatment.

In summary, our data show that protein prenylation is essential for nuclear remodeling within the first 24-h period of iHL60 differentiation, independently from the differentiation process itself. Additionally, L-778,123 treatment exerts an effect on nuclear shape through a different mechanism than FTase- or GGTase I-mediated protein prenylation, as neither single nor double inhibition of both enzymatic activities reproduces the phenotype of the dual inhibitor.

### 3.6. Putative Involvement of Novel GGTase III in Nuclear Remodeling

To date, there are no reported off-targets of L-778,123. Nevertheless, a recently discovered enzymatic complex with prenyl-transferase activity dubbed GGTase III could be a novel target of this FTase/GGTase I dual-inhibitor. To explore this possibility, we performed a series of computational analysis based on molecular modeling. We used molecular docking to model L-778,123 binding to all known prenyl-transferases and report here the results of 100 runs of docking, with poses clustered at 2 Å Root Mean Square Deviation (RMSD) tolerance.

First, we assessed the reliability of our in silico approach by docking L-778,123 with known targets FTase and GGTase I, for which co-crystals are available and where inhibition mechanism requires coordination with the catalytic zinc ion in the active site. In both cases, molecular docking of L-778,123 successfully identifies a preferential pose consistent with the crystal structure in which the imidazole group coordinates with the zinc ion (Figure 6a). For FTase, 81% of docked structures cluster together with an average binding energy of −11.6 ± 0.2 kcal/mol, while for GGTase I the most numerous cluster comprises 75% of poses with an average binding energy of −11.1 ± 0.2 kcal/mol (Appendix A). Additionally, we considered GGTase II (RabGGTase) as a “negative control” given its proven resistance to inhibition by L-778,123. Molecular docking to the apo-enzyme structure with two superimposed PO_4_^3−^ anions in the pyrophosphate pocket fails to identify a pose coordinating the drug with the catalytic zinc ion, with the pose representative of 75% of docked structures occupying the middle of the active site (Figure 6b) at an average binding energy of −7.4 ± 0.1 kcal/mol (Appendix A).

According to our modeling, docking to the active site of GGTase III in complex with two PO_4_^3−^ anions is indeed possible. The analysis shows that 76% of poses coordinating the imidazole ring with the catalytic zinc cation similarly to FTase and GGTase I (Figure 6c), with an average binding energy of −7.8 ± 0.3 kcal/mol. Molecular docking failed with apo-GGTase III structure or complexed with geranylgeranyl pyrophosphate (GGPP) (Appendix A), suggesting inhibitory activity only in presence of anions in the pyrophosphate pocket consistent with GGTase I-inhibitory mechanism. In such conformation, the drug occupies the isoprenoid substrate pocket (Figure 6d).

In summary, using computational modeling we put forward the hypothesis of inhibitory activity of L-778,123 towards the newly discovered prenyl-transferase GGTase III and confirm the soundness of the in silico modeling by comparing results for previously known FTase, GGTase I and II. If confirmed by experimental evidence, this result would extend the application of L-778,123 as a triple-inhibitor of protein prenylation and provide an explanation for the lack of phenotypic overlap in iHL60 nuclear remodeling between L-778,123 and FTI-277/GGTI-298 treatments.

## 4. Discussion

### 4.1. The Role of Cytoskeleton in Nuclear Remodeling

The forces enacted by cytoskeletal elements are a key component of dynamic remodeling of nuclear morphologies during basic cellular processes. In adherent cells, actin microfilaments (MFs) pivot on focal adhesion points to deform the nucleus, organized in specialized structures like transmembrane actin-associated nuclear (TAN) lines or actin-caps [34,35,36] by actin-binding proteins controlling actomyosin function [13,14,37]. Similarly, microtubules (MTs) produce rapid nuclear shape fluctuations [38,39] and nuclear proximity of the centrosome briefly deforms the nuclear volume upon sudden MT-repolymerization [18].

Indeed, we show that nuclear morphology is deeply affected by the same forces even in adhesion-independent cell types, where MF-stabilization promotes “choked” nuclear morphologies reminiscent of actomyosin hypercontraction [15]. Equally, we reproduce compression and wrinkling of the nucleus observed in adherent cells following MT disassembly [40,41], linking this process directly to actomyosin hypercontraction promoted by MT-bound RhoA-activating factors [42,43,44]. Additionally, even low protein levels of vimentin intermediate filaments (IF), known for organizing in cage-like perinuclear structures [45,46], play a role in organelle-positioning [47], as their disruption lengthens relative distances of granulocyte nuclear lobules.

Nevertheless, the involvement of these “classical” forces in an array of more complex nuclear morphological features such as nuclear invaginations, nuclear tunnels and nuclear segmentation is still unclear [9]. While nuclear invaginations contain cytosolic cytoskeletal filaments [17] and aberrant MT-nucleation sites promote generation of similar structures, a causative link between the two in physiological conditions was not observed [19]. Similarly, mechanistic involvement of cytosolic cytoskeletal networks in generation and structural integrity maintenance of nuclear lobules appears limited. Both are independent from MFs [10,48,49], and this notion extends to IFs in our experiments in a system where only vimentin and keratin-10 (KRT10) are effectively present. Moreover, we show that lobular collapse observed upon MT-depolymerization [48,49] is purely driven by hyper-activation of actomyosin contraction, with nuclear morphology unscathed even after 24 h of complete cytoskeletal depolymerization upon VNCT/Noco + LatA/CytD double treatments or inhibition of ROCK. Previously, a mechanistic link between MTs and lobule generation has been proposed in an apoptosis-resistant cell line over a 7-day culture period [49]. In our system, ~90% of segmented iHL60 nuclei are achieved over 3 days and careful titration of MT-depolymerizing drug shows that lobulation and segmentation are not inhibited by MT-depolymerization at concentrations functionally impairing cell and nuclear divisions. Additionally, lobulation and segmentation still proceed without MF- and MT-contributions after Day 1 of ATRA induction, yielding aberrant but clearly lobulated and segmented nuclei by Day 3.

### 4.2. The Role of Cholesterol Biosynthetic Pathway in Nuclear Remodeling

While downregulation of the cholesterol biosynthesis might be linked to immune system specification and terminal differentiation [50,51,52], several arguments prompted a more detailed investigation of this pathway. Firstly, mutations of one of its key enzymes (laminB-receptor, LBR) cause Pelger-Huët anomaly [53], a condition in which haploinsufficiency results in hypolobulated granulocytic nuclei [31], while excess gene copies promote hyperlobulation [30,32]. Secondly, pathways involved in lipid metabolism play preeminent roles in membrane bending and nuclear invagination genesis, with phosphatidylcholine-related enzyme CCTα being essential for their organization [54,55,56]. Additionally, pathological conditions like diabetes characterized by impairment of lipid metabolism induce nuclear deformation and even pseudo-lobulation in lymphocytes [57]. Thirdly, cholesterol and several up- and down-stream metabolic intermediates are potent signaling molecules [50,58,59,60,61] and two direct derivatives of the pathway (estradiol and progesterone) have been recently shown to regulate nuclear invagination in endometrial cells [62].

Indeed, pharmacological inhibition of several steps of cholesterol biosynthesis induces changes in nuclear shape. Treatment of undifferentiated HL60 cells with HMGCR and CYP51A1 inhibitors promotes, respectively, nuclear deformation (statins), and appearance of deep nuclear invaginations (ketoconazole) (Appendix A), possibly by inducing differentiation [63,64,65,66]. More interestingly, AY9944 impairs nuclear lobulation and segmentation in differentiating iHL60 at concentrations effective in inhibiting the sterol reductase activity of TM7SF2 and LBR [58,67]. LBR is a dual-function protein characterized by a nucleoplasmic domain with DNA-, lamin-, and chromatin organizing protein-binding sites and an enzymatically active membrane domain catalyzing the same reaction as TM7SF2 [68,69], but in an essential role in human cells [70,71]. Although its role in structuring the chromatin and binding of LMNB1 is most likely to affect the nuclear shape, the possibility of a functional cross-talk between the two domains is intriguing, especially in light of a limited ability of the sole reductase domain to rescue nuclear lobulation in a small percentage of murine ichthyotic neutrophils [72]. Alternatively, the metabolite-flow unbalance generated by depletion of LBR enzymatic activity might account for the observed effect on nuclear shape, with accumulation of two members of the meiosis-activating sterol family MAS-412 and FF-MAS and their direct nuclear signaling activity [58].

Our transcriptomic profiling shows that the isoprenoid branch of the pathway is finely tuned during differentiation, with an early targeted rebalancing of metabolic flows, much alike neuronal maturation [73]. Bisphosphonates are a classical strategy to impair protein prenylation by upstream substrate depletion, but HL60 are notoriously resistant to FDPS inhibition [74], and in our experiments only the highest sub-precipitation drug concentration of the most potent zoledronate induced polyploidization consistent with type 1 geranylgeranyl transferase (GGTase I) inhibition (Appendix A).

### 4.3. The Role of Prenyl-Transferases in Nuclear Remodeling

FTase inhibitors (FTIs) affect nuclear morphology predominantly by acting on farnesylated nuclear lamina components LMNA/C, LMNB1, and LMNB2. Prenylation of lamins is an essential post-translational modification regulating membrane biogenesis in several non-human models, where its impairment results in excessive NE growth, NE in-folding and nuclear lobulation [75,76,77], while inducing tubulation of both nuclear and synthetic membranes by direct bending of the lipid bilayer [78]. FTase inhibition in human cells generates donut-shaped nuclei in fibroblasts undergoing aberrant mitosis [79] and smoothing blebbed and deformed nuclei in Hutchinson-Gilford progeria cells [80]. In our system, FTI-treatment smoothens undifferentiated HL60 nuclei to almost perfectly spherical shapes, and impairs the assembly of DNA-containing and nuclear lamina-coated filaments connecting nuclear segments in induced iHL60. While the molecular structure of nuclear filaments is still undefined [29,81], it is not implausible that the underlying mechanism requires coordination of lamin- and heterochromatin-binding complexes driven by specific DNA sequences. To this point, in addition to lamins farnesyl moieties are attached to centromere-binding proteins [79], and centromeric and pericentromeric regions are bound by LBR during neutrophil differentiation and extensively relocalized to the nuclear periphery following chromosomal supercontraction [27]. Moreover, LBR coordinates other DNA binding proteins, including HP1α, MeCP2, and the farnesylated LMNB1, with this latter binding domain seemingly responsibly for the strongest lobulation rescue-capability in ichthyotic murine neutrophils [72]. Indeed, ectopic overexpression of LBR [82] or its ligand LMNB1 [83] induces dramatic nuclear in-folding and excess of nuclear invaginations. Thus, a mechanism for nuclear lobulation and segmentation employing DNA binding and chromatin condensation or pulling is reasonable, not unlike the condensin-mediated process generating nuclear invaginations in *D. melanogaster* polytene nuclei [24]. 

The surprising nuclear phenotype observed upon L-778,123-treatment clearly goes beyond inhibition of FTase or GGTase I. Although replicates FTase inhibition in undifferentiated HL60, no multinucleation, polyploidization, or retained lobulation are present in iHL60, while reproduced in double-inhibition experiments by individually specific drugs. The essential role in vesicle trafficking, docking and fusion of small prenylated proteins accounts for the surface marker absence in iHL60 and prenylation-inhibition of a Ras protein blunts signal transduction for respiratory burst activation [84]. Nevertheless, L-778,123 does not impair nuclear remodeling by preventing differentiation. Additionally, it is unlikely to exert its effect by structural impairment, as the effective-window of the drug is limited to the first 24 h of ATRA induction and dispensable at band-like cell stage. Thus far, no known off-targets of this clinically trialed drug have been reported [85,86,87], but the recent discovery of a new complex between PTAR1 and RABGGTB with geranylgeranyl transferase activity (GGTase III) [88,89] provides an exciting possibility to explain our results and significantly narrow the pool of protein candidates accountable for nuclear remodeling. This hypothesis is supported by molecular modeling for docking potential of L-778,123 to GGTase III active site. The proposed inhibition mechanism and the requirement of anions in the pyrophosphate pocket is strikingly similar to inhibition mechanism of GGTase I, where anions strongly enhance drug binding [87]. While the presented in silico data are compelling, with successful docking to FTase and GGTase I consistent with drug co-crystals [87] and failed docking to GGTase II consistent with absence of inhibitory activity on Rab geranylgeranylation [86], experimental evidence will be required to better understand the effect of L-778,123 on remodeling of nuclear morphology in human granulocytes. Importantly, at least in one of the only two GGTase III substrates reported thus far, efficiency of geranylgeranylation of SNARE protein Ykt6 is strictly dependent on pre-existing farnesylation [88], thus opening the possibility to an additional explanation of the effect of FTIs on nuclear morphology by impairment of GGTase III substrate-recognition.

## 5. Conclusions

In conclusion, in this study we advance the model of a nucleus-intrinsic mechanism for nuclear remodeling in human granulocytes by considering nuclear lobulation and segmentation as models of self-induced deformation. As shown by our experimental results, the contribution of cytosolic cytoskeleton in generating and maintaining such structures is limited. Granulocytic differentiation of HL60 cells does not require nuclear remodeling, thus these two events can be seen as concurrent and not causative, opening the chance of transferring the studied mechanism for nuclear segmentation to other nuclear features such as nuclear invaginations and tunnels. Sterol metabolism and protein prenylation play key roles in the morphological change, either by regulating intracellular molecular signaling or impairment of physical association of protein–lipid membrane complexes. Finally, if experimentally confirmed, we report for the first time a cellular effect of pharmacological inhibition of a newly discovered transferase, GGTase III, by a clinically relevant anticancer drug previously employed for FTase and GGTase I inhibition. The identification of novel key players in nuclear shape remodeling not only provides insight into molecular mechanisms of cell biology, but can have broad therapeutic applications for cancer, in which morphological changes to the nucleus play important roles in malignancy and correlate with poor prognosis.

## Figures and Tables

**Figure 1 cells-09-02509-f001:**
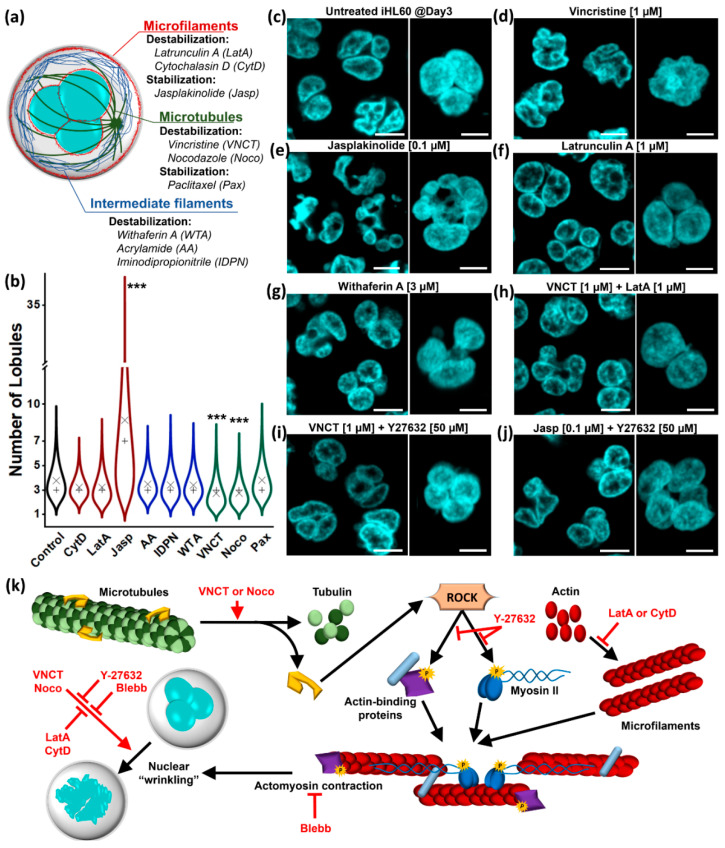
Effect of cytoskeleton on lobular structure maintenance. (**a**) Schematic representation of cytosolic cytoskeleton networks and component-specific small molecules (presented with effect on target component and employed drug abbreviation). (**b**) Quantification of lobule number in live-cell DNA staining in Day3 iHL60 (minimum sample number = 100; × = mean; + = median; *** = *p* < 0.001, Kruskal–Wallis test with Dunn post-hoc comparison). (**c**–**j**) Representative pictures of live-cell nuclear staining in Day 3 iHL60 treated for 3 h with the reported small molecule or combination of molecules (mid-focal section scale bar = 10 μm; 3D reconstruction inset scale bar = 5 μm). (**k**) Schematic representation of the complex interactions between MTs and MFs relative to the nuclear “wrinkling” effect, highlighting how cytosolic cytoskeletal components can be specifically, but not independently, targeted.

**Figure 2 cells-09-02509-f002:**
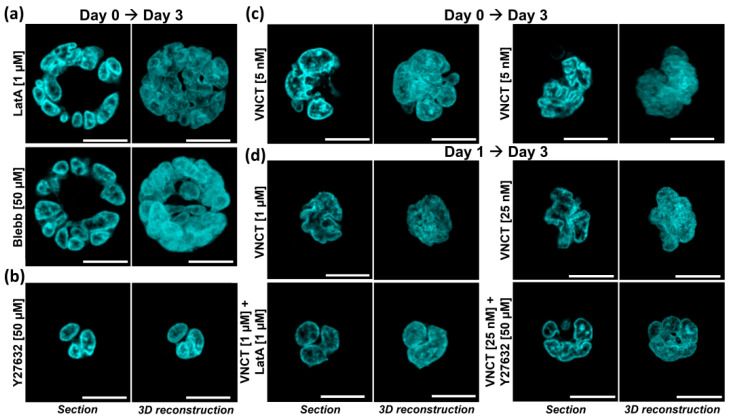
Effect of cytoskeleton on lobulation and segmentation. (**a**) Representative pictures of live-cell nuclear staining in Day 3 iHL60 treated for 3 days with LatA or Blebb, showing giant cells with multiple lobulated and segmented nuclei. (**b**) Representative picture of live-cell nuclear staining in Day 3 iHL60 treated for 3 days with ROCK inhibitor Y-27632. (**c**) Nanomolar concentrations of vincristine (VNCT) are 97% lethal when added at Day 0 of differentiation, but surviving cells display giant polyploid nuclei, either lobulated and segmented (left) or presenting a “wrinkled” morphology (right). (**d**) iHL60 treated at Day 1 are more resistant to microtubules (MTs)-depolymerization given early cell-cycle exit, and simultaneous microfilament (MF)-disassembly (left) or ROCK inhibition (right) rescue nuclear wrinkling and allow nuclear lobulation and segmentation. (All images up-to-scale for comparison, scale bar = 10 μm).

**Figure 3 cells-09-02509-f003:**
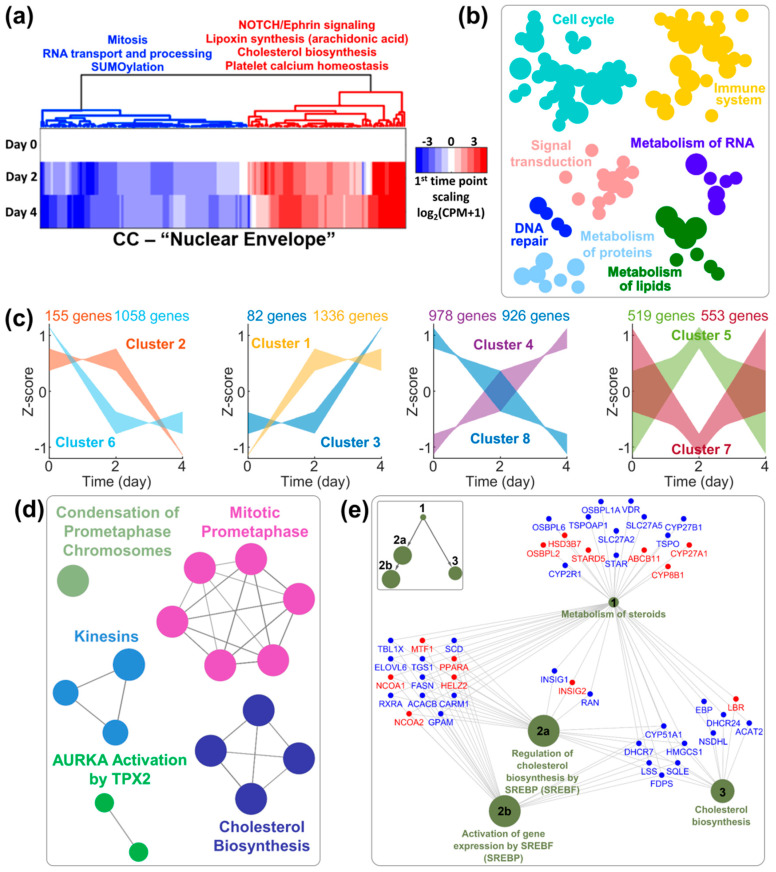
Transcriptomic profiling of HL60 differentiation. (**a**) Differentially expressed genes (DEGs) in the “Nuclear Envelope” Cellular Component category clustered according to expression pattern, with main Reactome terms for cluster enrichment analyses. (**b**) Reactome categories for enrichment analyses of all DEGs between Day 4 and Day 0 (enrichment analysis, B-H post-test, *p* < 0.05). (**c**) DEG clustering according to temporal expression profile at the three considered time-points. (**d**) Reactome terms (clustered according to k-score) for late downregulated DEGs (cluster 2 in (**c**)) (enrichment analysis, B-H post-test, *p* < 0.01). (**e**) Gene network of dark-blue cluster shown in (**d**) showing downregulation (blue) of the majority of DEGs in the “Cholesterol biosynthesis” node between Day 0 and Day 4, with only laminB-receptor (LBR) upregulated (red). Inset: Hierarchical Reactome structure.

**Figure 4 cells-09-02509-f004:**
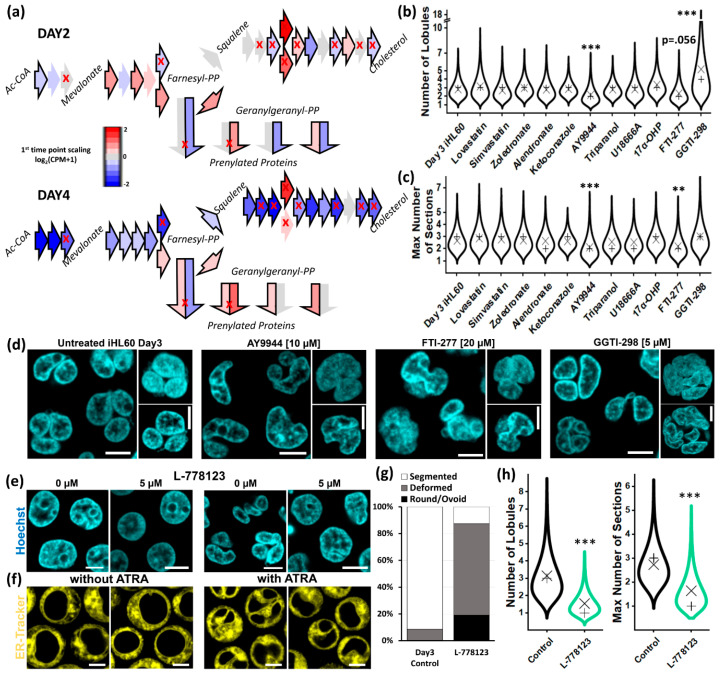
Effect of cholesterol pathway inhibitors on lobulation. (**a**) Cholesterol biosynthesis pathway, shown as sequence of enzymatic reactions with each arrow representing the responsible enzyme color-coded according to gene expression level against Day 0 (bold border = DEG at *p* < 0.001; red cross = druggable target (see Appendix A); double arrows are enzymatic complexes with two subunits). (**b**,**c**) Quantification of number of lobules per nucleus (minimum number of quantified cells per condition = 83) and maximum number of sections (minimum number of quantified cells per condition = 91) in iHL60 differentiated in presence of the indicated drug (× = mean; + = median; ** = *p* < 0.01, *** = *p* < 0.001, Kruskal–Wallis test with Dunn post hoc comparison). (**d**) Representative pictures of live-cell nuclear staining in Day 3 iHL60 after inhibition of TM7SF2/LBR enzymatic activity and protein prenylation, with aberrant nuclear morphology when compared to vehicle-treated iHL60 (insets are 3D reconstructions of a single representative nucleus and corresponding mid-section, scale bar = 5 μm). (**e**,**f**) Dual FTase/GGTase I inhibitor L-778,123 exerts the most dramatic effect in nuclear morphology, smoothening the nuclei of HL60 cells (left) and abolishing both nuclear lobulation and segmentation in iHL60 cells (right) (scale bar = 5 μm). The effect of L-778,123 can be quantified by (**g**) qualitative description of nuclear morphology and (**h**) quantification of number of lobules (minimum sample number = 79) and maximal number of nuclear sections (minimum sample number = 53) at Day 3 (× = mean; + = median; *** = *p* < 0.001, Kruskal–Wallis test with Dunn post-hoc comparison).

**Figure 5 cells-09-02509-f005:**
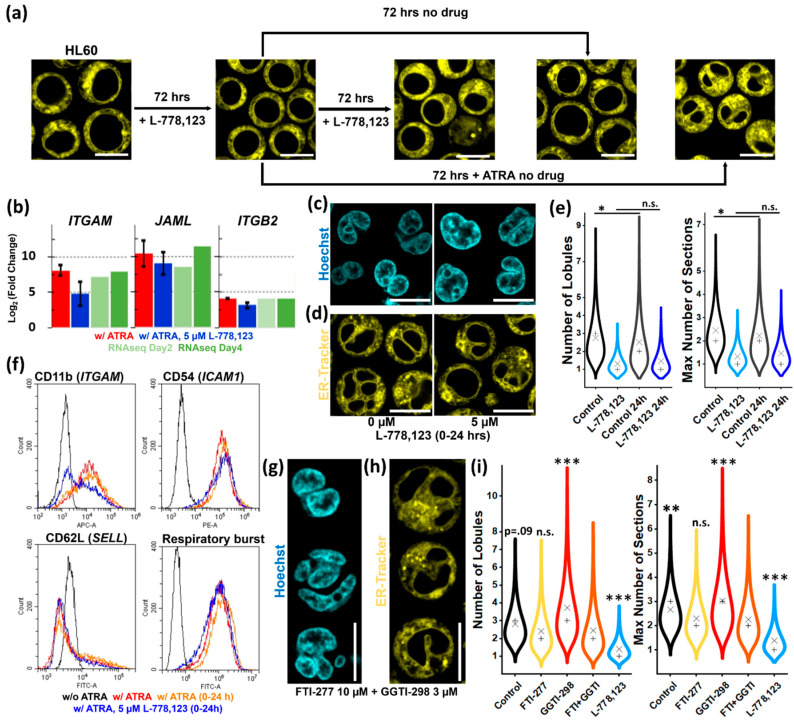
Effect of protein prenylation inhibition on HL60 cells. (**a**) Live endoplasmic reticulum (ER) staining of HL60 following treatment as indicated (scale bar = 10 μm). (**b**) Expression of granulocytic differentiation markers in Day 3 iHL60 with or without L-778,123 treatment (*n* = 3), compared with expression levels from RNAseq data. (**c**,**d**) Representative pictures of live-cell imaging showing the effect of differentiation induction with 24-h all-trans-retinoic acid (ATRA) treatment with or without 24-h L-778,123 treatment at Day 3 (scale bar = 10 μm). (**e**) Quantification of number of lobules (minimum sample number = 157) and maximal number of nuclear sections (minimum sample number = 146) for the experiment shown in (**c**,**d**) (× = mean; + = median; * = *p* < 0.05, n.s. = not significant, all other comparisons are *p* < 0.001, Kruskal–Wallis test with Dunn post-hoc comparison). (**f**) Cytofluorimetric analyses for granulocyte-specific surface markers in Day 3 iHL60 following 24-h ATRA treatment with or without L-778,123. (**g**,**h**) Representative pictures of live-cell imaging after double farnesyltransferase (FTase) and type 1 geranylgeranyltransferase (GGTase I) inhibition with enzyme-specific drugs, which does not reproduce the effect of dual-inhibitor L-778,123 in Day 3 iHL60 (scale bar = 10 μm). (**i**) Quantification of number of lobules (minimum sample number = 106) and maximal number of nuclear sections (minimum sample number = 110) for the experiment shown in (**g**,**h**) (× = mean; + = median; all comparisons are relative to “FTI + GGTI” sample, *** = *p* < 0.001, ** = *p* < 0.01, n.s. = not significant, Kruskal–Wallis test with Dunn post-hoc comparison).

**Figure 6 cells-09-02509-f006:**
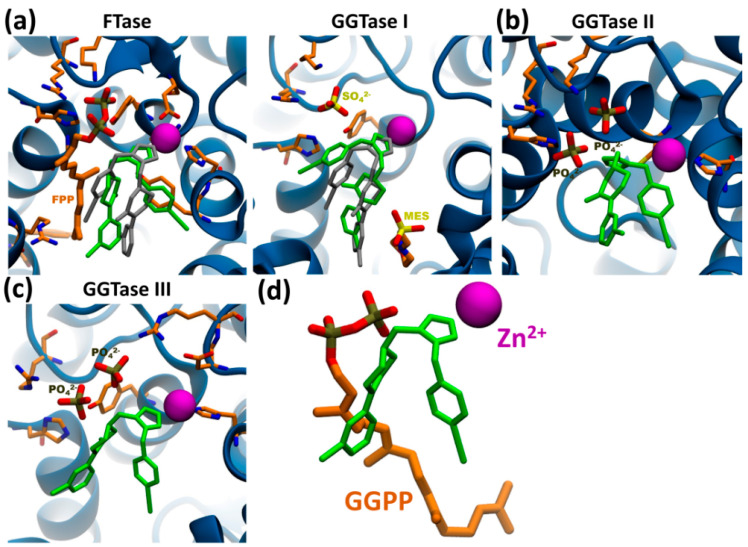
Docking of L-778,123 to prenyl-transferase active sites. (**a**) Docking of L-778,123 to crystal structures of FTase (left, PDB: 1S63) and GGTase I (right, PDB: 1S64). The preferred conformations of the drug in green are consistent with the superimposed crystallized conformation in grey. L-778,123 inhibitory activity requires chelation of the catalytic zinc ion (purple) through coordination of the imidazole ring. Important amino acid side chains in orange. (**b**) A preferred conformation for L-778,123 docking to GGTase II (PDB: 3DSS with superimposed two PO_4_^3-^ ions) is possible in presence of anions in the pyrophosphate pocket (see Appendix A) and does not coordinate with the catalytic zinc ion. (**c**) L-778,123 docks to GGTase III (PDB: 6J74) co-crystalized with anions in the pyrophosphate pocket (see Appendix A), with the imidazole group coordinating with the catalytic zinc ion, similarly to known targets FTase and GGTase I in (**a**). (**d**) Superimposition of docked L-778,123 structure from (**c**) with GGPP within GGTase III (PDB: 6J7X) shows the probable mechanism of inhibition by occupying the isoprenoid pocket in the active site.

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
