# Peer review of "Nuclear Morphological Remodeling in Human Granulocytes Is Linked to Prenylation Independently from Cytoskeleton"

_cells, 2020, doi:10.3390/cells9112509_

Round 1
Reviewer 1 Report
The paper describes the nuclear describes the changes in nuclear morphology associated with metamyelocytes differentiation into granulocytes. Lobulations of nuclei can be modulated by the cytoskeleton but upon suppression of actin or MT lobulations remain suggesting a nucleus “intrinsic” mechanism. While the paper does overstate this fact, I do believe the general conclusion and that this is interesting. During differentiation using ATRA the paper finds specific and variable changes in transcription, the most interesting being downregulation of cholesterol biosynthesis except LBR. Targeting enzymatic steps revealed parts of this pathway contributes to nuclear shape. Protein prenylation was found to be one of the most significantly important, but while it suppressed nuclear lobulation it did not change differentiation markers. Finally the paper postulates GGTase III could be the major prenyl-transferase candidate being inhibited by L-778,123. Overall, this paper will be a great interest to the field. Upon changes to bring graphs and data into the main figures of the paper, I would suggest publication in the journal Cells.
Many graphs of nuclear lobulation are hidden in the supplemental figure. Figure S1A graph, some data in Figure S6, and Figure S15A graph MUST be moved to the main text for clearer data presentation. Simply providing a single Hoechst image is not sufficient. I also thought that Figure S18 was very nice. The authors should consider what other data hiding it the supplemental might be better suited in the main figures.
In figures showing both Hoechst and ER-Tracker a clear distinction should be make that they are not images of the same cells. Currently it appears as if images of the same cells were taken with two different fluorescent markers, which is not the case. I would suggest putting a black line between the images or labeling them as different panels.
The paper might consider a small discussion of the fact that chromatin compaction/mechanics via chromatin proteins or post-translational modifications also is known to determine nuclear morphology. The discussion of lamins was void of the fact that secondary effects to chromatin and/or its attachment to lamins may be the reason for nuclear shape changes.
Author Response
We thank the reviewer for the comments and suggestions. We revised the manuscript according to these suggestions and improved the quality of the paper. Attached here is the revised version of the manuscript and all edits are highlighted in yellow. The manuscript now contains revised Main Figures 1, 4 and 5, and Supplementary Figures S3 and S10.
Below are responses to the specific points raised by the reviewer.
Many graphs of nuclear lobulation are hidden in the supplemental figure. Figure S1A graph, some data in Figure S6, and Figure S15A graph MUST be moved to the main text for clearer data presentation. Simply providing a single Hoechst image is not sufficient. I also thought that Figure S18 was very nice. The authors should consider what other data hiding it the supplemental might be better suited in the main figures.
According to the reviewer’s suggestion we moved several figures from Supplementary Materials to main figures. Both Fig. S15 and Fig. S18 have been edited in the revised version of the manuscript into Fig. 4B,C and Fig.5A, respectively. The rescue experiments for Jasplakinolide are now reported in Fig. 1J instead of Fig. S3A.
Due to space constraints for effective figure presentation, we decided to retain Fig. S1A as supplementary, as morphometric quantification graphs are paired with the presented images (“lobule number” for nuclear staining, “max number of sections” for ER staining) and both Fig. 1B and Fig. S1A lead to the same conclusions. While in all other figures quantifications of both morphometric parameters are duly reported, in Fig. 1C-J presentation of nuclear staining provides a more immediate understanding of the nuclear morphologies observed and quantified in Fig. 1B.
To this point, we revised the data presentation for cholesterol biosynthesis pathway in Fig. 4, providing quantifications for both morphometric parameters in the main figure and giving more space to presenting the aberrant nuclear morphologies in iHL60 cells induced by the effective drugs.
In figures showing both Hoechst and ER-Tracker a clear distinction should be make that they are not images of the same cells. Currently it appears as if images of the same cells were taken with two different fluorescent markers, which is not the case. I would suggest putting a black line between the images or labeling them as different panels.
The distinction has been made clear in the revised version of Fig. 4 and Fig. 5 following the reviewer’s suggestion.
The paper might consider a small discussion of the fact that chromatin compaction/mechanics via chromatin proteins or post-translational modifications also is known to determine nuclear morphology. The discussion of lamins was void of the fact that secondary effects to chromatin and/or its attachment to lamins may be the reason for nuclear shape changes.
The Discussion section has been revised and edited for more clarity and the involvement of lamins and DNA binding has been highlighted in a new paragraph.

Reviewer 2 Report
Nuclear shape changes are associated with differentiation and behavior of many cell types. It is generally assumed that cytoplasmic components, especially the cytoskeleton play a major role in shaping the nuclei, given their tight connection to the nuclear envelope via the LINC complex, for example. Although the cytoskeletal contribution is certainly valid, the role of nuclear factors and especially the nuclear envelope have been less investigated. Granulocyte differentiation is a prominent model for nuclear shape changes, given their striking nuclear morphologies.
The authors employ this model in vitro to analyze the contributions from the cytoskeleton and nuclear factors for the nuclear shape changes.
The authors convincingly reveal a function of the sterol pathway and especially prenylation but less so from the cytoplasm for the nuclear shape changes. Furthermore, by comparing the effect of inhibitors for farnesylation and geranyl-geranylation enzymes, they reveal with a computational approach a recently identified geranyl-geranyl-transferase (GGTase III) as a new player.
In the first part, the authors test a comprehensive list of drugs interfering with the cytoskeleton. Although functions for the nuclear morphology are observed, in no case the prominent nuclear lobulations are lost, leading to the conclusion that other factors, i. e. nuclear components, are at the center of nuclear shape changes. With transcriptional profiling during differentiation, several groups of genes are identified with changing expression during differentiation. They focus on the cholesterol/prenyl biosynthesis pathways by employing chemical inhibitors. The strongest effects were observed with inhibitors of the farnesyl transferase and geranyl-geranyl transferase. An inhibitor with a broad specificity gave the stronger effect that a combination of the FTI and GGTI leading to the identification of a new GGTaseIII in nuclear shape changes.
The claims of the abstract are properly supported by the presented data. Given the conclusiveness of the comprehensive and high quality data in the main figures and the long list of supplemental materials, the proper interpretation of the data and the timeliness of the research topic, I strongly recommend publication of the manuscript.
In the following I list a few minor issues, which the authors may address prior to publication.
- l305-306 ..neither affected nuclear…..
- Fig. 4 The authors may add the quantification of the FTI to panel F next to the results of L-778123 inhibition.
- l400 use unspecific instead of aspecific
- the Discussion is rather long. The authors may wonder to focus it on the specific topic and findings. What is missing however is a detailed discussion of the effects of prenylated proteins of the nuclear envelope and their effect on nuclear morphology/shape. A few of the previous papers are cited. Relating their findings to the previous literature could be more complete. There are number of studies about farnesylated IMN proteins, lamins and lamin-related proteins. It would be interesting for the reader to get the novel data discussed together with the previous papers. There are reports in the literature for farnesylated IMN proteins related but different from lamins and studies about the effect of farnesylated proteins on the structure of the nuclear envelope. The studies coming to my mind are from Drosophila, i. e. kugelkern which is farnesylated similar to lamins and a recent paper about the effect of farnesylated proteins on the nucleus in stem cells and enterocytes of the mid gut.
Author Response
We thank the reviewer for the comments and suggestions. We revised the manuscript according to these suggestions and improved the quality of the paper. Attached here is the revised version of the manuscript and all edits are highlighted in yellow. The manuscript now contains revised Main Figures 1, 4 and 5, and Supplementary Figures S3 and S10.
Below are responses to the specific points raised by the reviewer.
In the following I list a few minor issues, which the authors may address prior to publication.
l305-306 ..neither affected nuclear…..
Edited
Fig. 4 The authors may add the quantification of the FTI to panel F next to the results of L-778123 inhibition.
We revised both Fig. 4 and Fig.5 in the new version of the manuscript. Results of quantifications for samples treated with FTI-277 and L-778,123 are reported together and compared in Figure 5I, specific to results of all prenylation-inhibitor experiments.
l400 use unspecific instead of aspecific
Edited
the Discussion is rather long. The authors may wonder to focus it on the specific topic and findings. What is missing however is a detailed discussion of the effects of prenylated proteins of the nuclear envelope and their effect on nuclear morphology/shape. A few of the previous papers are cited. Relating their findings to the previous literature could be more complete. There are number of studies about farnesylated IMN proteins, lamins and lamin-related proteins. It would be interesting for the reader to get the novel data discussed together with the previous papers. There are reports in the literature for farnesylated IMN proteins related but different from lamins and studies about the effect of farnesylated proteins on the structure of the nuclear envelope. The studies coming to my mind are from Drosophila, i. e. kugelkern which is farnesylated similar to lamins and a recent paper about the effect of farnesylated proteins on the nucleus in stem cells and enterocytes of the mid gut.
We acknowledge the length and depth of the discussion. For purposes of clarity, the revised version of the manuscript has a slightly shorter discussion, divided into the three major points of research in order to focus the attention to each point in an orderly manner.
The involvement of lamins and DNA binding has been highlighted in a new paragraph, adding the requested reference.

Reviewer 3 Report
The results presented by Martewicz et al provide an exhaustive analyses of induced HL60 cells as a model of neutrophil differentiation upon treatment with multiple inhibitors of cytoskeletal protein assembly, thereby allowing the researchers to carefully assess how each inhibitor (or combinations of inhibitors) affect nuclear segmentation, a hallmark feature of mature neutrophils. The authors carefully step through effects of inhibiting microfilaments, microtubules and intermediate filaments, which in several interesting cases caused a disruption to segmentation leading to a “wrinkled” phenotype, or the termed “choked” phenotype indicating hyperlobulated nuclei. The results indicate that segmentation is independent of cytoskeletal forces, but nonetheless disrupting the cytoskeletal dynamics can dramatically influence the resulting patterns of nuclear lobulation. Interestingly, the phenotypes that are caused by agents that affect microtubules could be rescued by inhibitors of actomysin contraction, indicating actomysin hypercontraction might be a mediating mechanism in the “wrinkled” or “choked” phenotypes upon disruption of cytoskeletal components.
The authors then switch to transcriptomic profiling of the induced HL60 cells, which revealed the multiple downregulated factors to include multiple enzymes in cholesterol biosynthesis, which are opposite the expression pattern of the dual-function protein LBR. The authors then investigate the effects of inhibitors at multiple steps of cholesterol biosynthesis, and present perhaps the most interesting finding that the combination inhibitor L-778,123 caused a dramatic loss of nuclear lobulation. These results with other inhibitors indicated that disruption at the protein prenylation step of the isoprenoid branch of cholesterol biosynthesis alone can disrupt nuclear lobulation, thereby supporting the role of this function by LBR. The results provide new insight into not only the role of cytoskeletal proteins in nuclear lobulation but also how enzymatic processes in cholesterol biosynthesis might influence nuclear structural changes.
There are a few issues that could be considered as follows:
1. Lines 179-182, the authors show in the primary figures rescue of the "wrinkled" phenotype by MF depolymerization, but then place the rescue of "choked" morphologies caused by Jasp in the Supplemental figures - one might argue these data should also be included in Fig 1 as this is a very interesting observation (e.g., rescue of two very different abnormal morphologies, indicated the important role of MF hypercontraction).
2. Line 209 - 211, this sentence is confusing - the phrase "viability of iHL60 differentiated following MT-disruption is ~3% above 1.25 nM" assume indicates that only 3% of cells survive when HL60 cells are induced while treated with concentrations of VNCT above 1.25 nM, so perhaps reword (as written in the Fig 2 legend, which is clearer).
3. Lines 311- 314, the authors go from using inhibitors of FTase or GGTase I, which cause opposing effects on lobulation, to the dual inhibitor "L-778,123", but provide no explanation of this combination inhibitor with regards to the first two inhibitors. This is important for the reader to understand, as the most dramatic affects, complete abrogation of segmentation, is caused by the dual inhibitor. A brief description of the dual inhibitor would be helpful.
4. Line 357-358, the authors show that multiple neutrophil-specific markers are expressed by the iHL60 cells when treated with L-778,123 as shown in Fig 5A, and then they transition to the data in Fig S19 that indicates the "treated iHL60 failed to confirm granulocytic lineage". As written this is confusing and contradicts the data shown in Fig 5 - all markers indicate that the treated, induced cells are indeed showing hallmark molecular changes and the respiratory burst, but that is not consistent with S19B and C - how are these two experiments different with regards to iHL60 treatments with L-778,123? This needs to be more clearly described to clear up any potential confusion.
5. Lines 500-513, the authors discuss how their results with AY9944 with impaired nuclear segmentation correlates to phenotype associated with loss of LBR or TM7SF2, but they did not mention past studies that specifically focus on the sterol reductase activities of LBR vs. TM7SF2, which may shed some light on their results – specifically how the sterol reductase domain alone can rescue hypolobulation due to loss of LBR, and that lack of TM7SF2 fails to show the same phenotypes (and cannot rescue those associated with loss of LBR). These points could be briefly discussed, see Subramanian et al, J Immunol. 2012 Jan 1;188(1):85-102; Bennati et al, FEBS J. 2008 Oct;275(20):5034-47, and Bennati et al, Biochim Biophys Acta. 2006 Jul;1761(7):677-85.
6. Are they any other neutrophil functions affected by L-778,123 or the other drugs that suppress nuclear lobulation (other than the shown studies of the respiratory burst responses)? What about chemotaxis - if the cells are hypolobulated, then this should be affected. Some mention of additional phenotypes expected or that should be tested with the different inhibitors of the isoprenoid branch of cholesterol biosynthesis could be provided.
Minor issues:
- Line 97, the number of cells for Total RNA isolation may be incorrect, as its indicated to be 107 (10e7 is more likely).
- Line 304 to 305, "did not affect neither....nor" is a double negative, might change to "had no effect on either... or..." or "neither... nor.. were affected".
- In Figure 4, it would helpful to indicate where in cholesterol biosynthesis pathway the sterol reductase activities of LBR or TM7SF2 interact - this would help in understanding the overlap of protein prenylation as affected by the inhibitors vs. phenotypes resulting from mutations of LBR.
Author Response
We thank the reviewer for the comments and suggestions. We revised the manuscript according to these suggestions and improved the quality of the paper. Attached here is the revised version of the manuscript and all edits are highlighted in yellow. The manuscript now contains revised Main Figures 1, 4 and 5, and Supplementary Figures S3 and S10.
Below are responses to the specific points raised by the reviewer.
There are a few issues that could be considered as follows:
Lines 179-182, the authors show in the primary figures rescue of the "wrinkled" phenotype by MF depolymerization, but then place the rescue of "choked" morphologies caused by Jasp in the Supplemental figures - one might argue these data should also be included in Fig 1 as this is a very interesting observation (e.g., rescue of two very different abnormal morphologies, indicated the important role of MF hypercontraction).
We agree with the reviewer that some of the data presented originally as Supplementary should be included in the main figures. The revised version of Figure 1, rescue images for both Vincristine and Jasplakinolide are presented relative to Y-27632 double treatment, while blebbistatin is presented in Fig. S3.
Moreover, Figure 4 and 5 have been also edited to include data previously reported as supplementary.
Line 209 - 211, this sentence is confusing - the phrase "viability of iHL60 differentiated following MT-disruption is ~3% above 1.25 nM" assume indicates that only 3% of cells survive when HL60 cells are induced while treated with concentrations of VNCT above 1.25 nM, so perhaps reword (as written in the Fig 2 legend, which is clearer).
Edited
Lines 311- 314, the authors go from using inhibitors of FTase or GGTase I, which cause opposing effects on lobulation, to the dual inhibitor "L-778,123", but provide no explanation of this combination inhibitor with regards to the first two inhibitors. This is important for the reader to understand, as the most dramatic affects, complete abrogation of segmentation, is caused by the dual inhibitor. A brief description of the dual inhibitor would be helpful.
Edited
Line 357-358, the authors show that multiple neutrophil-specific markers are expressed by the iHL60 cells when treated with L-778,123 as shown in Fig 5A, and then they transition to the data in Fig S19 that indicates the "treated iHL60 failed to confirm granulocytic lineage". As written this is confusing and contradicts the data shown in Fig 5 - all markers indicate that the treated, induced cells are indeed showing hallmark molecular changes and the respiratory burst, but that is not consistent with S19B and C - how are these two experiments different with regards to iHL60 treatments with L-778,123? This needs to be more clearly described to clear up any potential confusion.
Edited
Lines 500-513, the authors discuss how their results with AY9944 with impaired nuclear segmentation correlates to phenotype associated with loss of LBR or TM7SF2, but they did not mention past studies that specifically focus on the sterol reductase activities of LBR vs. TM7SF2, which may shed some light on their results – specifically how the sterol reductase domain alone can rescue hypolobulation due to loss of LBR, and that lack of TM7SF2 fails to show the same phenotypes (and cannot rescue those associated with loss of LBR). These points could be briefly discussed, see Subramanian et al, J Immunol. 2012 Jan 1;188(1):85-102; Bennati et al, FEBS J. 2008 Oct;275(20):5034-47, and Bennati et al, Biochim Biophys Acta. 2006 Jul;1761(7):677-85.
The Discussion section has been revised and edited for more clarity and the studies suggested by the reviewer have been appropriately discussed and referenced in the text.
Are they any other neutrophil functions affected by L-778,123 or the other drugs that suppress nuclear lobulation (other than the shown studies of the respiratory burst responses)? What about chemotaxis - if the cells are hypolobulated, then this should be affected. Some mention of additional phenotypes expected or that should be tested with the different inhibitors of the isoprenoid branch of cholesterol biosynthesis could be provided.
While the impact of prenylation inhibition on neutrophil functional features could be investigated, in this study we focused our attention solely in nuclear morphology. The results of marker expression, antigen presentation and respiratory burst are presented as proof of retained activation of neutrophil-differentiation program under L-778,123 treatment. At the present time, we did not investigate other functional properties of hypolobulated iHL60 after L-778,123 treatment.
Minor issues:
Line 97, the number of cells for Total RNA isolation may be incorrect, as its indicated to be 107 (10e7 is more likely).
Edited
Line 304 to 305, "did not affect neither....nor" is a double negative, might change to "had no effect on either... or..." or "neither... nor.. were affected".
Edited
In Figure 4, it would helpful to indicate where in cholesterol biosynthesis pathway the sterol reductase activities of LBR or TM7SF2 interact - this would help in understanding the overlap of protein prenylation as affected by the inhibitors vs. phenotypes resulting from mutations of LBR.
Fig. S10, that previously identified arrows in Fig. 4A with their corresponding gene names, has been edited in the revised version of the manuscript to highlight the enzymatic steps whose inhibition affects nuclear morphology.

Round 2
Reviewer 1 Report
The authors have addressed my minor revisions. The paper is ready for publication.
Reviewer 2 Report
The manuscript is now suitable for publication.
I have no further criticism.
Reviewer 3 Report
The authors have addressed this reviewer's concerns and improved the quality of the manuscript.